# Endoplasmic Reticulum Stress Causing Apoptosis in a Mouse Model of an Ischemic Spinal Cord Injury

**DOI:** 10.3390/ijms24021307

**Published:** 2023-01-09

**Authors:** Kiran Kumar Soni, Jinsu Hwang, Mahesh Ramalingam, Choonghyo Kim, Byeong C. Kim, Han-Seong Jeong, Sujeong Jang

**Affiliations:** 1Department of Physiology, Chonnam National University Medical School, Hwasun 58128, Republic of Korea; 2Department of Neurosurgery, Kangwon National University School of Medicine, Chuncheon 24341, Republic of Korea; 3Department of Neurology, Chonnam National University Hospital, Chonnam National University Medical School, Gwangju 61469, Republic of Korea

**Keywords:** apoptosis, endoplasmic reticulum stress, free radical, spinal cord injury

## Abstract

A spinal cord injury (SCI) is the devastating trauma associated with functional deterioration due to apoptosis. Most laboratory SCI models are generated by a direct impact on an animal’s spinal cord; however, our model does not involve the direct impact on the spinal cord. Instead, we use a clamp compression to create an ischemia in the descending aortas of mice. Following the success of inducing an ischemic SCI (ISCI), we hypothesized that this model may show apoptosis via an endoplasmic reticulum (ER) stress pathway. This apoptosis by the ER stress pathway is enhanced by the inducible nitric oxide synthase (iNOS). The ER is used for the protein folding in the cell. When the protein folding capacity is overloaded, the condition is termed the ER stress and is characterized by the accumulation of misfolded proteins inside the ER lumen. The unfolded protein response (UPR) signaling pathways that deal with the ER stress response then become activated. This UPR activates the three signal pathways that are regulated by the inositol-requiring enzyme 1α (IRE1α), the activating transcription factor 6 (ATF6), and the protein kinase RNA-like ER kinase (PERK). IRE1α and PERK are associated with the expression of the apoptotic proteins. Apoptosis caused by an ISCI is assessed using the terminal deoxynucleotidyl transferase dUTP nick end labeling (TUNEL) test. An ISCI also reduces synaptophysin and the neuronal nuclear protein (NeuN) in the spinal cord. In conclusion, an ISCI increases the ER stress proteins, resulting in apoptosis in neuronal cells in the spinal cord.

## 1. Introduction

An ischemic spinal cord injury (ISCI) is a complication after a thoracoabdominal aortic aneurysm, which causes injury to the nerves and leads to sensory and motor deficits, as well as widespread disabilities [1]. An ISCI seriously affects the quality of life in approximately 5% to 10% of patients [2]. Different studies have disclosed the pathophysiology of an ISCI, which includes the neuronal inflammation, oxidative stress, reactive changes in the glia, neuronal degeneration, and apoptosis [3]. The SCI pathology is divided in two stages: the first is caused by a direct effect on the spinal cord, and the second is caused by inflammation, the vascular dysfunction, autophagy, oxidative stress, and neuronal apoptosis [4,5,6]. Free radicals are produced after a SCI [7]. The inducible nitric oxide synthase (iNOS) expressed in the cells is responsible for the damage and inflammation. It contributes to a high production of nitric oxide (NO), which leads to the formation of free radicals and ultimately to apoptosis [8]. A SCI causes a reduction in the neuronal markers, such as the neuronal nuclear protein (NeuN) and synaptic markers, such as synaptophysin [9,10]. Following a SCI, some of the cells die, due to the post-traumatic necrosis, whereas other neuronal cells die by apoptosis [11]. We are more focused on apoptosis in our current experiment. Apoptosis is a programmed cell death that is increased under pathological conditions, it can also take place in physiological conditions [12]. During apoptosis, there is shrinkage of the cell membrane forming bleb and the formation of double membraned autolysosomes during autophagy. This is different from necroptosis, where there is swelling of the cell membrane [13,14]. Neurons and oligodendrocytes both show an apoptotic cell death [15]. Apoptosis inhibits the nerve function following a SCI [16]. It has been demonstrated that endoplasmic reticulum (ER) stress encourages apoptosis [17]. The ER is a cellular organelle found throughout the cytoplasm of all eukaryotic cells [18]. During a SCI, an unfolded protein response (UPR) occurs, due to a large number of misfolding proteins, caused by the changes in the microenvironment in the injured tissue [9]. This UPR activates three signal pathways that are regulated by the inositol-requiring enzyme 1α (IRE1α), activating the transcription factor 6 (ATF6), and the protein kinase RNA-like ER kinase (PERK) [19,20]. IRE1α and PERK show the expression of apoptotic proteins and it leads to the neuronal death [21]. A time course experiment in rats showed that apoptosis occurred, as early as 4 h post-injury and could be seen in decreasing amounts, as late as 3 weeks after a SCI [22]. However, this type of SCI is performed in rats, and most of the models are made by contusion injuries. We are first to show an ER stress resulting in apoptosis in a mouse model of an ISCI prepared by the occlusion of the descending aorta. This model mimics a patient undergoing cardiovascular surgery with paralysis, following the surgical procedure. In this mouse model, the seizures and paralysis are seen approximately 40–48 h postoperatively. This is the reason we did not look for apoptosis 24–48 h postoperatively. The purpose of this study is to evaluate the extent of apoptosis caused by an ER stress in the spinal cord of ISCI mice. 

## 2. Results

### 2.1. ISCI-Associated Changes in Locomotion

All of the ISCI animals had noticeable paralysis of the hind limbs, compared with the sham mice. The sham animals had higher BMS scores than the ISCI mice. The ISCI animals with completely paralyzed hind limbs had BMS scores of zero, compared with other ISCI animals that were not completely paralyzed (Figure 1).

### 2.2. ISCI-Associated Changes in the PERK Pathways

The quantities of the major proteins related to the ER stress were sharply affected by an ISCI. The glucose-regulated protein-78 (GRP-78) is an important marker of the ER stress [20]. The level of the GRP-78 protein was significantly higher in mice in the ISCI 1 week group and the ISCI 2 week group, than in the sham animals. The increased markers of the ER stress-related proteins lead to elevated phosphorylated PERK and phosphorylated IRE, which enhance the cascade of steps leading to apoptosis. With a higher GRP-78 protein level, the pPERK was significantly elevated in the ISCI 2 week animals, compared with the sham animals, and the ISCI 1 week animals showed higher pPERK levels, but the difference was not statistically significant. pPERK leads to an increase in the phosphorylated eukaryotic initiation factor 2 (peIF2) protein in the ISCI 1 week and ISCI 2 week animals, compared with the sham animals. The total form of eIF2 was used as a control, similar to beta-actin. The graphical analysis was performed with (*p*/t)eIF2. The increased peIF2 protein levels led to the increased activating transcription factor 4 (ATF4) protein levels. The ATF4 levels were significantly higher in the ISCI 2 week but not in the ISCI 1 week animals, than in the sham animals (Figure 2). The ATF4 leads to the production of the ER stress-induced apoptosis protein C/EBP homologous protein (CHOP). The details of the CHOP results are explained in a separate section.

### 2.3. ISCI-Associated Changes in the IRE Pathways

As mentioned earlier, GRP-78 leads to two pathways. Among those, the phosphorylated IRE also leads to apoptosis. pIRE was elevated in the ISCI 1 week and ISCI 2 week animals, compared with the sham animals. This leads to the elevation of the phosphorylated apoptosis signal-regulating kinase (pASK). The pASK was significantly higher in the ISCI 2 weeks group than in the sham group, whereas the pASK did not show a significant increase in the ISCI 1 week animals. The pASK leads to two subdivisions of the pathways, known as the phosphorylated P38 mitogen-activated protein kinase (pP38MAPK) and the phosphorylated c-Jun N-terminal kinases (pJNK). The pJNK was significantly higher in the ISCI 1 week and the ISCI 2 week animals than in the sham animals. The total form of JNK was used as a control, similar to beta-actin. The graphical analysis was performed with (*p*/t)JNK. The pP38MAPK was significantly elevated in the ISCI 1 week and ISCI 2 week animals, compared with the sham animals (Figure 3). The pP38MAPK and pJNK lead to the production of the mitochondrial induced apoptosis protein Bax. The details of the Bax results are explained in the following section. 

### 2.4. Western Blot and IHC Showing the Elevation of Apoptosis in an ISCI

The ATF4 from the PERK pathway leads to the production of the CHOP, followed by the (p53 upregulated modulator of apoptosis) PUMA. The CHOP and PUMA were significantly elevated in the ISCI 1 week and ISCI 2 week animals, compared with the sham animals (Figure 4a). There was also a significant increase in the chromogenic (DAB) staining of the CHOP antibody in the ISCI 1 week and ISCI 2 week animals, compared with the sham animals. Considering the sham with 100 cells, the ISCI 1 week and 2 week mice had 240 ± 35.38 and 286.46 ± 22.65 more CHOP positive cells than the sham mice (Figure 4b). The pP38 MAPK and pJNK from the IRE pathway lead to the production of the pro-apoptotic protein Bax. Bax was elevated in the ISCI 1 week and ISCI 2 week animals, compared with the sham animals. In addition to the pro-apoptotic protein PUMA and Bax, we also examined the cleaved caspase 3 as an important protein leading to apoptosis. The cleaved caspase 3 was also significantly elevated in the ISCI 1 week and ISCI 2 week animals, compared with the sham animals.

### 2.5. ISCI-Associated Elevation of the Free Radical Proteins

The SCI led to the significant elevation of iNOS in the ISCI 1 week and ISCI 2 week animals, compared with the sham animals (Figure 5). 

### 2.6. Western Blot and IHC Showing the Reduction of the Neuronal Markers in an ISCI

The SCI in mice led to the significant reduction of the neuronal markers NeuN and synaptophysin in the ISCI 1 week and ISCI 2 week animals, compared with the sham animals (Figure 6a). The IHC also showed that the neuronal markers, including NeuN and synaptophysin, were significantly reduced in the ISCI 1 week and ISCI 2 week animals, compared with the sham animals (Figure 6b,c). The NeuN cells in the ISCI 1 week and 2 week mice were 28.63 ± 3.07 and 30.43 ± 2.54, in comparison with the 100 NeuN cells counted in the sham mice. Similarly, the synaptophysin density in the ISCI 1 week and 2 week mice were 38.79 ± 5.13 and 28.13 ± 3.07, respectively, compared to the 100 cells in the sham mice. 

### 2.7. IHC Showing the Elevation of Apoptosis in an ISCI

The CHOP is an important marker for apoptosis in the ER stress pathway. In addition to the western blot performed with the spinal cord tissue, the animals perfused with PFA were used for the IHC. The IHC staining with TUNEL of the spinal cord also showed significant numbers of apoptotic cells in the ISCI 1 week and ISCI 2 week animals, compared with the sham animals (Figure 7). The apoptotic positive cells in the ISCI 1 week and 2 week mice were 1205.23 ± 373.44 and 1632.36 ± 498.55, respectively, in comparison with the 100 apoptotic cells in the sham mice. 

## 3. Discussion

In this study, we established a SCI model by occluding the descending aortas of mice, which is different from the traditional SCI animal models that are formed by a direct impact on the spinal cord (contusions, transection, etc.). Our model mimics a clinical SCI caused by thoracoabdominal surgery [23]. A benefit of our model is that the spinal cord was not morphologically affected, while there were massive histological changes in the spinal cord [24]. An ISCI in thoracoabdominal aortic surgery is caused by the disproportionate amount of oxygen required and low oxygen delivery produced by an aortic occlusion [23]. 

The locomotion is affected by the SCI in mice. This result was similar to those of previous SCI studies, which included either a SCI induced by the occlusion of the aorta, for a brief amount of time, or a SCI induced by contusions in mice [24,25]. 

Previous studies have provided ample evidence that a SCI causes the death of a large number of neurons and the nerve conduction disruption, due to the loss of synapses [9,26]. Among them, motor function has a direct impact on the loss of neurons [27]. Our study also demonstrated a reduction in synaptophysin and NeuN. 

Following the induction of a SCI in animals, the nerve cells are triggered by the protein folding errors, which are also called unfolded protein aggregation, inside the ER [28]. This condition is known as the ER stress. ER stress can stimulate a sequence of changes, and the misfolded proteins piled up in the ER are processed; this process of misfolded proteins is then called the unfolded protein response (UPR) [29,30,31]. The accumulation of the misfolded proteins is the main stimulus for apoptosis induced by ER stress via the intrinsic pathway [32]. The intrinsic stimuli for apoptosis are ER stress, DNA damage, hypoxia, and metabolic stress, whereas the extrinsic stimulus for apoptosis are the death cell receptors or lethal receptors [32,33]. The unfolded proteins upregulate the expression of chaperones, such as GRP-78. GRP-78 then instigates the activation of the pPERK and pIRE signaling pathways [34,35].

When ER stress occurs, there is oligomerization and phosphorylation of the PERK, which leads to the phosphorylation of Eif-2 and the activation of the *p*-eIF2 pathway [36]. Phosphorylation of eIF2 further upregulates ATF-4, which targets the activation of the apoptotic protein CHOP [32,37,38]. The CHOP activates either of the pro-apoptotic BH3 only proteins, such as Bim, Bid, Bad, Bmf, Noxa, and PUMA. These proteins are able to bind and activate the direct effectors Bak and Bax [39].

Another signaling pathway that is activated by GRP-78 is pIRE. This is the most studied activation mechanism [40]. IRE1 activates the inflammatory response and the cellular apoptosis-associated protein kinases, especially the apoptosis signal regulating kinase 1 (ASK1), which leads to the phosphorylation of two pathways, including p38 MAPK and JNK. The phosphorylated p38 MAPK and JNK further leads to the activation of the pro-apoptotic protein Bak and Bax [32,41,42]. Bak and Bax, once activated, form an oligomer and eventually make pores in the outer mitochondrial membrane. This leads to the permeability of the outer mitochondria [37,39,43]. Then, the pro-apoptotic proteins are released from the inner membrane to cytosol, that initiates the caspase activation which subsequently results in apoptosis [39]. There is sufficient evidence to indicate that the ER-stress-activated pathways play a significant role in apoptosis in a SCI [9,44].

The pathological processes following a SCI occur due to oxidative stress, such as the formation of the reactive oxygen species (ROS), reactive nitrogen species, and the free radicals, as well as inflammation [45,46]. The main source of NO is iNOS [46], and it is an important marker for the ROS-induced free radicals [47]. ER stress and oxidative stress are bidirectionally related, and the aggregation of the unfolded proteins in the ER lumen is enough to trigger the production of the ROS [47,48,49].

If the ER protein homeostasis is not restored and there is excessive oxidative stress, apoptotic cell death may occur via the upregulation of the CHOP, PUMA, Bax, followed by the cleaved caspase 3 [47,50,51].

There are some pharmacological drugs that makes the modulation in PERK and IRE pathway, that eventually leads to a therapeutic benefit. These drugs are rapamycin, AMPK activators, dipeptidyl peptidase IV (DPP-IV) inhibitors, and angiotensin II type 1 receptor blockers (ARBs) [52]. Rapamycin weakens the apoptosis induced by ER stress via the suppression of the IRE1-JNK signaling pathway [53]. The AMPK activators attenuate the phosphorylation of the eukaryotic initiation factor 2 α (eIF2α) and the JNK signaling pathways [54]. The DPP-IV inhibitors protect against the ER stress-induced apoptosis and inflammation through the IRE1α/JNK-p38 and PERK/CHOP -mediated pathways [55]. ARBs attenuate the ER stress mediated apoptosis by the suppression of the glucose-regulated protein 78 (GRP78), ATF4, and CHOP [56]. 

In our research, we demonstrated that the ER stress induced apoptosis, in response to an ISCI and the levels of proteins that activate apoptosis, including the GRP78, CHOP, PUMA, Bax, and the cleaved caspase-3, were significantly higher in mice with an ISCI, based on the western blot and IHC. A TUNEL assay also showed the elevated numbers of apoptotic cells in an ISCI. As intermediary signaling molecules associated with the ER stress pathway, these proteins mediate the apoptosis through different pathways, such as the IRE-ASK1-p38MAPK- pJNK-Bax- cleaved caspase-3, and the PERK-elF2a- ATF4-CHOP-PUMA-Bax pathways. A schematic diagram of an ISCI–ER stress–oxidative stress–apoptosis is shown in (Figure 8).

## 4. Materials and Methods

### 4.1. Animal Preparation

The study was approved by the Institutional Animal Care and Use Committee CNU IACUC-H-2022-36 of Chonnam National University, Jeollanam-do, Republic of Korea. Twenty-two female C57BL/6 mice were ordered from Damool, Daejeon, Republic of Korea (weight, 21–22 g; age, 12–13 weeks), were fed standard mice chow, and had free access to water; they were maintained in an animal facility under constant environmental conditions (room temperature, 20 ± 2 °C; relative humidity, 50 ± 10%; and a 12-h light-dark cycle) for a week before the surgical procedure was performed. 

### 4.2. Experimental Design

The mice were divided into three groups:ShamISCI 1 weekISCI 2 weeks

### 4.3. Animal Surgery

Each mouse was injected with 50 μL of sodium pentobarbital (JW Pharmaceuticals, Seoul, Republic of Korea), followed by gas anesthesia with isoflurane (Troikaa Pharmaceuticals Ltd., Gujarat, India) and oxygen. Artificial tears (EcolinR ophthalmic ointment, Santen Pharmaceuticals Co. Ltd., Osaka Japan) were used to maintain the moisture of the eyes. A midline incision was made near the neck to visualize the trachea, to insert an endotracheal tube for artificial ventilation, while the mouse was in the supine position. The left thoracic region was shaved and cleaned. A horizontal incision was made to expose the apex of the lungs. The window was enlarged using a separator. The descending aorta was exposed. A clamp was used to occlude the descending aorta for 8 min, to induce an ISCI. Then, the clamp was removed. The tissue and skin were sutured using Cat#W9981 (Ethicon, NJ, USA) and Cat#SK617 (Ailee Co. Ltd., Busan, Republic of Korea), respectively. Each mouse was maintained on oxygen without isoflurane, to regain conscious. Antibiotics were administered topically (Fucidine, Leopharma, Ballerup, Denmark) and subcutaneously (Kuhnil Pharmaceuticals, Seoul, Republic of Korea). Each mouse was continuously monitored for 40–48 h for seizures, followed by paralysis. Complete paralysis of the hind limbs was considered to be the best model of an ISCI. The surgical method was followed according to Awad et al. [21].

### 4.4. Locomotor Testing

The Basso, Beattie, Bresnahan locomotor rating scale (BBB) is used for rats, whereas the Baso Mouse Scale (BMS) is used as a locomotor rating scale for mice. Locomotor recovery following a SCI was identified by the scoring of the hind limb performance in an open field arena, using the BMS scoring system. Under blinded conditions, the locomotion score was determined within a period of 4 min. It ranges from 0–9, where 0 denotes no ankle movement and 9 denotes frequent stepping [57]. The BMS was evaluated for seven days.

### 4.5. Perfusion and Tissue Collection

One week after surgery in the ISCI 1 week group and two weeks after surgery in the ISCI 2 weeks and the sham surgery groups, the mice were injected with 100 μL of sodium pentobarbital and were perfused transcardially, first with 10 mL of 0.9% sodium chloride (Cat#CBS006A, LPS Solution Co. Ltd., Daejeon, Republic of Korea), followed by 50 mL of phosphate-buffered saline (PBS) (Cat#CBP007B, LPS Solution Co. Ltd., Daejeon, Republic of Korea) or 4% paraformaldehyde (PFA) (Cat#BPP9004, Chuncheon Bioindustry Foundation, Gangwon, Republic of Korea). The spinal cords from the mice perfused with PBS were removed and stored at −80 °C for the western blot. The spinal cords from the mice perfused with PFA were removed and post-fixed in the same fixative for 24 h. Then, they were placed in a sucrose solution for the immunohistochemistry (IHC). 

### 4.6. Western Blot

The tissues perfused with PBS were taken for the western blot. A tissue lysate buffer supplemented with protease, and phosphatase inhibitors were added and sonicated, and the solution was incubated for 30 min on ice. Thereafter, the tissue was centrifuged at 13,200 rpm for 15 min at 4 °C. The protein concentrations were determined using the BCA Protein Assay Kit (#23225, Thermo Scientific, Rockford, IL, USA) according to the manufacturer’s instructions. Thereafter, the samples were run on 6–10% SDS–polyacrylamide gels, then transferred onto polyvinylidene difluoride membranes (IPVH00010, Millipore, Bradford, MA, USA). The membranes were washed with PBS containing 0.5% (*v*/*v*) Tween 20 (PBS-T), blocked with a blocking buffer (5% nonfat dried milk solution or 5% bovine serum albumin (BSA) solution) prepared in PBS-T, and subsequently incubated overnight with primary antibodies, at 4 °C. The antibodies used are listed in (Appendix A). Thereafter, the membranes were washed with (PBS-T), and exposed to horseradish peroxidase-conjugated secondary antibodies, for 2 h at room temperature, and then washed three times with PBS-T. Finally, the signals were detected using an enhanced chemiluminescence (ECL) system (WBLUR0500, Millipore, Billerica, MA, USA) and a luminescent image analyzer (LAS 4000, GE Healthcare, Little Chalfont, UK). The membranes were incubated in Western Blot Stripping Buffer (Cat#21059, Thermo Scientific) with constant shaking for 60 min. It was blocked, and a subsequent primary antibody was used to detect other required proteins. The signals were visualized using the ECL system. Beta-actin was used to normalize the expression levels of the target proteins. The densitometric analysis was performed using ImageJ software (National Institutes of Health, Bethesda, MD, USA).

### 4.7. Histology

For the histological estimations, the tissue from the sucrose solution was removed and wrapped in tissue paper for a few seconds. The blocks were made in a cryomold (Cat#K4557, Sungwon Medical Company, Chungbuk, Republic of Korea) with cryo-embedding medium (FSC-22, Leica Biosystems, IL, USA), and three spinal cords were arranged in parallel and kept on dry ice, to solidify. The tissue blocks were marked and stored at −80 °C until further use. Cryostat (Leica CM1860) was switched on with a mold base, and a brush remained in it to maintain the temperature at approximately −30 °C. The spinal cords were sectioned coronally in 20 parallel series of 15-μm sections on the charged slides (Cat# 12-550-15, Fisher Scientific). The tissues adhered to the slides were allowed to remain at room temperature overnight and were then stored in a slide box at −20 °C.

### 4.8. IHC

The adhered tissues on the slides stored at −20 °C, were first thawed to room temperature. The sections were washed with PBS between all incubations. They were treated with 3% H_2_0_2_ for 15 min, followed by 0.5% TritonX-100, and normal goat serum. The sections were incubated overnight with primary antibodies.

The slides were washed on day 2 and treated with the corresponding biotinated secondary antibodies, followed by avidin-binding complex (ABC)-elite (Vector; Cat# PK-6100; 1:1000 in PBS; 30 min). The sections were next incubated in a diaminobenzidine (DAB) substrate kit (Cat# 34002, ThermoFisher Scientific) until the color of the tissue turned brown.

The slides were then washed with distilled water, followed by a series of dehydrations and cover slipping.

### 4.9. Terminal Deoxynucleotidyl Transferase dUTP Nick End Labeling (TUNEL) Assay

The DeadEnd™ Colorimetric TUNEL System (G7131 and G7332; Promega Corporation, Fitchburg, WI, USA) was used to quantify apoptosis in the spinal cord tissue, according to the manufacturer’s instructions. The dark brown cells were considered apoptotic cells.

### 4.10. Image Acquisition and Analysis

Once the slides were prepared, a Zeiss Axio Vert. A1 microscope was used to take images. The images were taken at magnifications of 5x and 20x for each section. All of the slides were chromogenically stained; thus, the dark brown sections in the image were considered positive cells. The analysis was performed using ImageJ software; the whole area of the section and the sections with positive cells were read. The area of the positive cells in the sham, ISCI 1 week and ISCI 2 week groups were calculated.

### 4.11. Statistical Analysis

All data are expressed as the mean ± standard error of the mean (SEM) of three independent experiments. The data processing was performed using Microsoft Excel, and SigmaPlot 12.0 (Systat Software, San Jose, CA, USA) was used to analyze the data and plot the graphs. The statistical significance of the ISCI groups was determined using a one-way analysis of variance, followed by Tukey’s post hoc multiple-comparison test. *p* < 0.05 was considered statistically significant.

## 5. Conclusions

In conclusion, our research demonstrated that a SCI can also be caused by the occlusion of the descending aorta during thoracoabdominal surgery. An SCI model prepared with this method also causes apoptosis and the loss of nerve cells in the spinal cords of mice via the ER stress pathways and oxidative stress. 

## Figures and Tables

**Figure 1 ijms-24-01307-f001:**
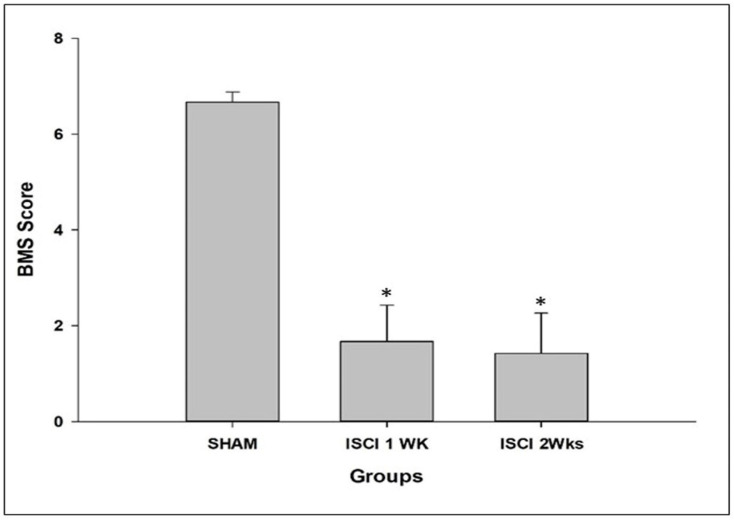
BMS score of the mice, post-surgery. Data are expressed as mean ± standard error. * *p* < 0.05 versus the sham group.

**Figure 2 ijms-24-01307-f002:**
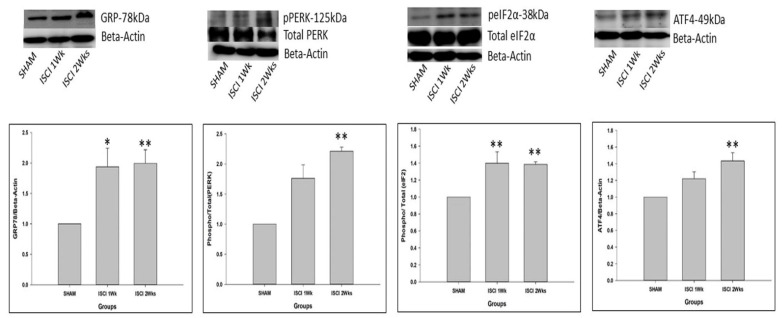
Western blot images with the quantification results showing the endoplasmic reticulum stress proteins by the PERK pathway. Data are expressed as mean ± standard error. * *p* < 0.05 versus the sham group; ** *p* < 0.01 versus the sham group.

**Figure 3 ijms-24-01307-f003:**
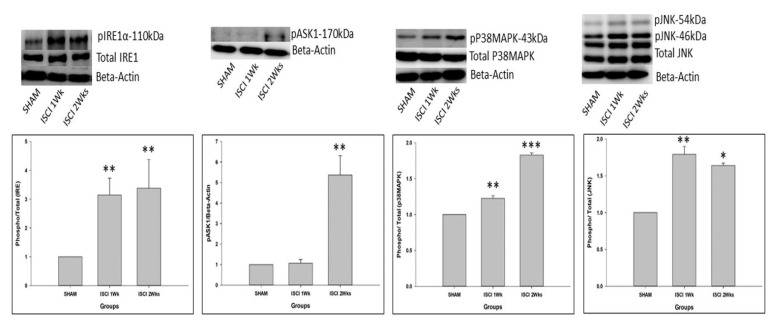
Western blot images with the quantification results showing the endoplasmic reticulum stress proteins by the IRE pathway. Data are expressed as mean ± standard error. * *p* < 0.05 versus the sham group; ** *p* < 0.01 versus the sham group; *** *p* < 0.001 versus the sham group.

**Figure 4 ijms-24-01307-f004:**
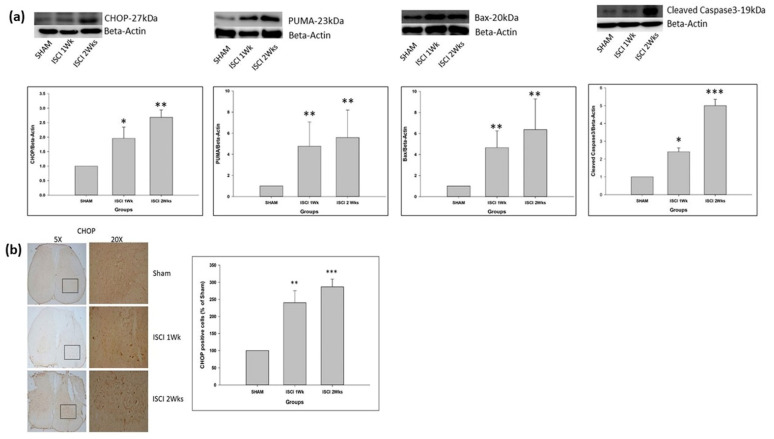
(**a**) Western blot images with the quantification results showing the CHOP, PUMA, Bax, and cleaved caspase 3; (**b**) Immunohistochemistry images with the quantification results showing the CHOP positive cells. Data are expressed as mean ± standard error. * *p* < 0.05 versus the sham group; ** *p* < 0.01 versus the sham group; *** *p* < 0.001 versus the sham group.

**Figure 5 ijms-24-01307-f005:**
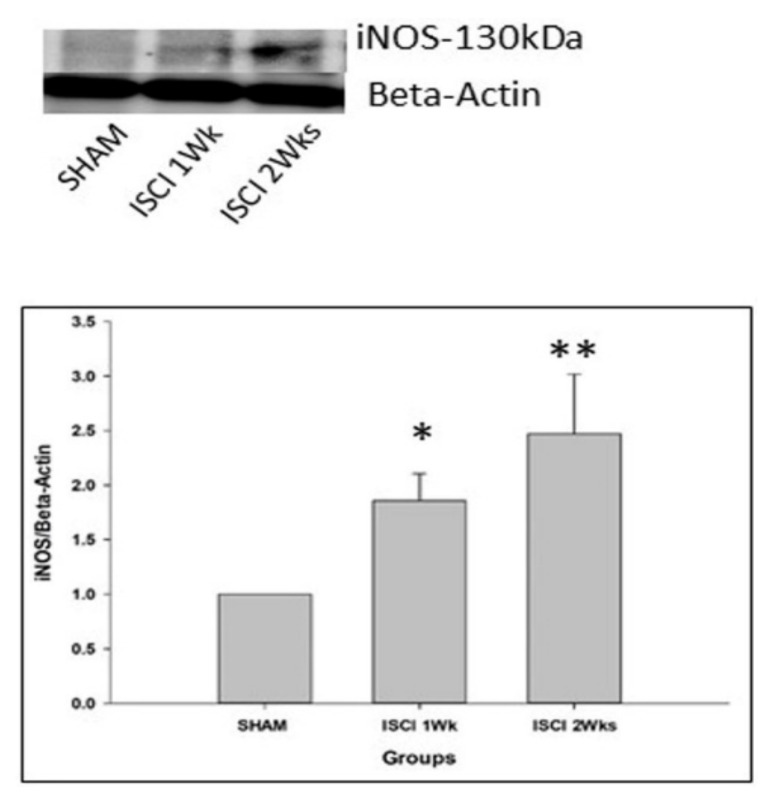
Western blot images with the quantification results showing the iNOS, marker of the free radical. Data are expressed as mean ± standard error. * *p* < 0.05 versus the sham group; ** *p* < 0.01 versus the sham group.

**Figure 6 ijms-24-01307-f006:**
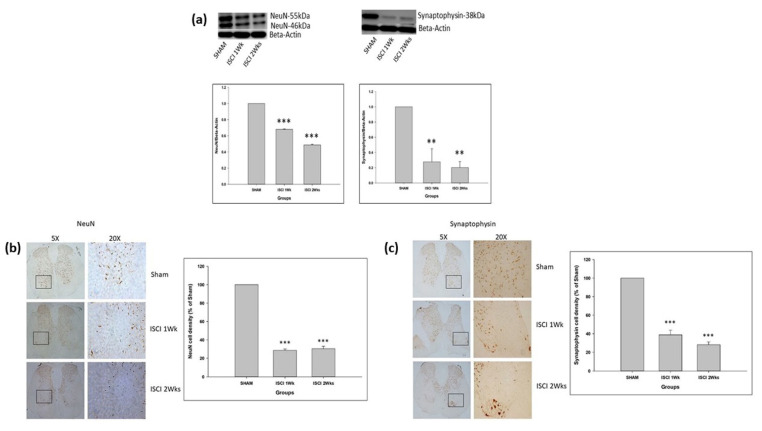
(**a**) Western blot images with the quantification results showing NeuN and synaptophysin; (**b**) Immunohistochemistry images with the quantification results showing the NeuN positive cells; (**c**) Immunohistochemistry images with the quantification results showing the synaptophysin positive cells. Data are expressed as mean ± standard error. ** *p* < 0.01 versus the sham group; *** *p* < 0.001 versus the sham group.

**Figure 7 ijms-24-01307-f007:**
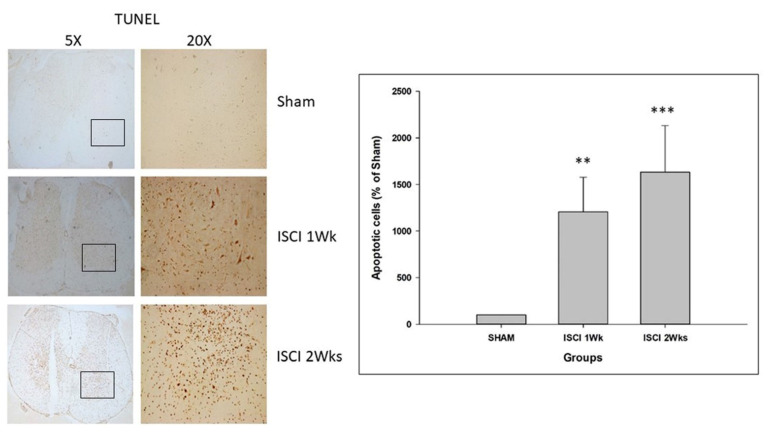
Immunohistochemistry images with the quantification results showing the TUNEL positive cells. Data are expressed as mean ± standard error. ** *p* < 0.01 versus the sham group; *** *p* < 0.001 versus the sham group.

**Figure 8 ijms-24-01307-f008:**
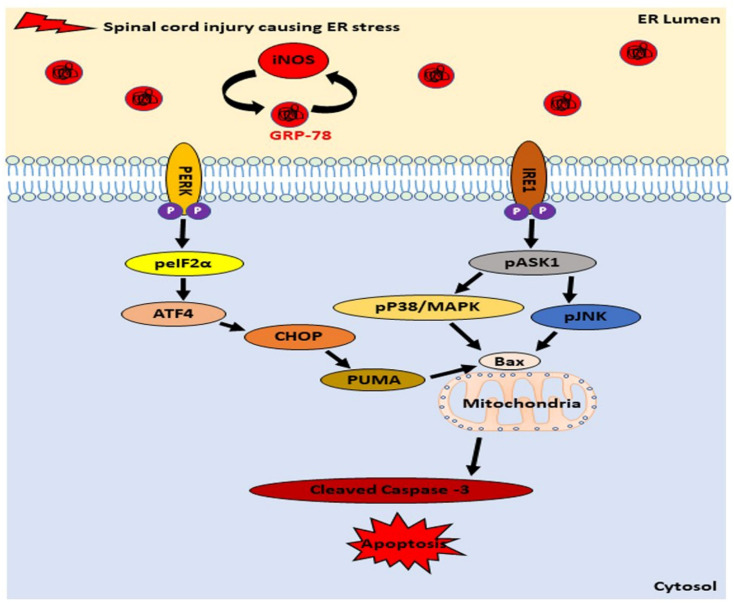
Schematic diagram showing the relation between an ischemic spinal cord injury, iNOS, and endoplasmic reticulum stress proteins by the PERK and IRE pathway leading to apoptosis.

## Data Availability

The data presented in this study are available in article and Appendix A.

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
