# Peer review of "Endoplasmic Reticulum Stress Causing Apoptosis in a Mouse Model of an Ischemic Spinal Cord Injury"

_ijms, 2023, doi:10.3390/ijms24021307_

Round 1
Reviewer 1 Report
Discussion could be more in depth regarding the ER stress induced signaling and consequences.
1. ER stress can lead to activation of either the intrinsic (mitochondrial) or extrinsic death receptor pathways of apoptosis. Which pathway this paper described?
2. Did the authors consider analyzing TRAIL , particularly TRAIL-R1/DR4 and TRAIL-R2/DR5, since CHOP pathway can elevate the expression of these proapoptotic genes which promote extrinsic apoptosis. Apart from CHOP, the BH3-members (PUMA, NOXA and BIM) are important components of ER stress-induced apoptosis as well.
3. ER stress can induce not only apoptosis, but also mediate nonapoptotic cell death such necroptosis and autophagy cell death. Can the authors results distinguish the difference between these types of cell death?
4. Based on the authors findings, are there pharmacological modulators of PERK , IRE1 or ATF 4 that can bring therapeutic benefit?
Author Response
Que:1. ER stress can lead to activation of either the intrinsic (mitochondrial) or extrinsic death receptor pathways of apoptosis. Which pathway this paper described?
Ans: Thank you for your detailed review and valuable comments.
We have added sentences in track changed version line 203-206 with their respective references.
The accumulation of misfolded proteins is the main stimulus for apoptosis induced by ER stress via intrinsic pathway [32]. Intrinsic stimuli for apoptosis are ER stress, DNA damage, hypoxia and metabolic stress, whereas extrinsic stimulus for apoptosis are death cell receptors or lethal receptors [32,33]
Que:2. Did the authors consider analyzing TRAIL, particularly TRAIL-R1/DR4 and TRAIL-R2/DR5, since CHOP pathway can elevate the expression of these proapoptotic genes which promote extrinsic apoptosis. Apart from CHOP, the BH3-members (PUMA, NOXA and BIM) are important components of ER stress-induced apoptosis as well.
Ans: We have not analyzed TRIAL, since we have added sentences according to Reviewer 1, Question 1, “The accumulation of misfolded proteins is the main stimulus for apoptosis induced by ER stress via intrinsic pathway”. We have not analyzed extrinsic pathway.
We have added sentences in track changed version line 212-214, with their respective references.
CHOP activates either of pro-apoptotic BH3 only proteins such as Bim, Bid, Bad, Bmf, Noxa and PUMA. These proteins are able to bind and activate the direct effectors Bak and Bax [39].
We have added sentences in track changed version line 219-225, with their respective references.
Phosphorylated p38 MAPK and JNK further leads to activation of the pro-apoptotic protein Bak and Bax [32,41,42]. Bak and Bax once activated forms an oligomer and eventually make pores in the outer mitochondrial membrane this leads to permeability of outer mitochondria [37,39,43]. Afterwards, pro-apoptotic proteins are released from inner membrane to cytosol that initiates caspase activation which subsequently results to apoptosis[39]. There is sufficient evidence to indicate that indicate that ER-stress-activated pathways play a significant role in apoptosis in SCI [9,44].
Que:3. ER stress can induce not only apoptosis, but also mediate nonapoptotic cell death such necroptosis and autophagy cell death. Can the authors results distinguish the difference between these types of cell death?
Ans: We have added sentences in track changed version line 49-52 with their respective references.
, it can also take place in physiological conditions [12]. During apoptosis is shrinkage of cell membrane forming bleb which is different from necroptosis where there is swelling of cell membrane whereas there is formation of double membraned autolysosomes during autophagy [13,14].
Que:4. Based on the authors findings, are there pharmacological modulators of PERK, IRE1 or ATF 4 that can bring therapeutic benefit?
Ans: We have added sentences in track changed version line 235-243 with their respective references.
There are some pharmacological drugs that makes modulation in PERK and IRE pathway that eventually leads to therapeutic benefit. These drugs are rapamycin, AMPK Activators, Dipeptidyl Peptidase IV (DPP-IV) Inhibitors and Angiotensin II Type 1 Receptor Blockers (ARBs) [52]. Rapamycin weakens apoptosis induced by ER stress via suppression of the IRE1-JNK signaling pathway [53]. AMPK Activators attenuates phosphorylation of eukaryotic initiation factor 2 α (eIF2α) and JNK signaling pathways [54]. (DPP-IV) Inhibitors protects against ER stress-induced apoptosis and inflammation through IRE1α/JNK-p38 and PERK/CHOP -mediated pathways [55]. (ARBs) attenuates ER stress mediated apoptosis by suppression of glucose-regulated protein 78 (GRP78), ATF4, and CHOP [56].

Reviewer 2 Report
Soni and Jiang et.,al showed an ER stress associated cell death in ischemic spinal cord injury mouse model. The authors states that the ISCI leads to activation of UPR pathway which further activates the downstream signaling cascades and causes neuronal death. The study was well done however, the manuscript needs to be structured well and author need to perform some additional experiments to validate their conclusion. Here are some of the concerns:
1. Authors need to restructure the manuscript. Fig 4 and 7 needs to be combined and Fig 6, 9 and 10 needs to be combined as they are providing similar conclusions by two different techniques.
2. Did authors observed any abnormal protein aggregations in neurons or other non-neuronal cells in ISCI animals which could trigger the activation of UPR?
3. It would be very interesting for the authors to use drugs targeting the ER stress like tauroursodeoxychloic acid or salubrinal to rescue the phenotype observed in ISCI animals. This will further strengthen the conclusion that neuronal loss in ISCI animals is mainly via ER stress.
Author Response
Soni and Jiang et.,al showed an ER stress associated cell death in ischemic spinal cord injury mouse model. The authors states that the ISCI leads to activation of UPR pathway which further activates the downstream signaling cascades and causes neuronal death. The study was well done however, the manuscript needs to be structured well and author need to perform some additional experiments to validate their conclusion. Here are some of the concerns:
Que: 1. Authors need to restructure the manuscript. Fig 4 and 7 needs to be combined and Fig 6, 9 and 10 needs to be combined as they are providing similar conclusions by two different techniques.
Ans: Thank you for your detailed review and valuable comments.
We have revised the images accordingly.
Que:2. Did authors observed any abnormal protein aggregations in neurons or other non-neuronal cells in ISCI animals which could trigger the activation of UPR?
Ans: No we have not observed any abnormal protein aggregations in neurons or other non-neuronal cells in our experiment which could trigger the activation of UPR.
Que:3. It would be very interesting for the authors to use drugs targeting the ER stress like tauroursodeoxychloic acid or salubrinal to rescue the phenotype observed in ISCI animals. This will further strengthen the conclusion that neuronal loss in ISCI animals is mainly via ER stress.
Ans: In current experiment we have not used drugs that would target ER stress. We would use your valuable suggestions in future experiments.

Round 2
Reviewer 2 Report
The manuscript looks fine now.